# The Differential Prognostic Impact of Long-Duration Atrial High-Rate Episodes Detected by Cardiac Implantable Electronic Devices between Patients with and without a History of Atrial Fibrillation

**DOI:** 10.3390/jcm11061732

**Published:** 2022-03-21

**Authors:** Hironori Ishiguchi, Yasuhiro Yoshiga, Akihiko Shimizu, Takeshi Ueyama, Makoto Ono, Masakazu Fukuda, Takayoshi Kato, Shohei Fujii, Masahiro Hisaoka, Tomoyuki Uchida, Takuya Omuro, Takayuki Okamura, Shigeki Kobayashi, Masafumi Yano

**Affiliations:** 1Department of Medicine and Clinical Science, Division of Cardiology, Yamaguchi University Graduate School of Medicine, Ube 755-8508, Japan; yoshiga820@gmail.com (Y.Y.); ueyama23@yahoo.co.jp (T.U.); makoto.ono@ukyu.edu (M.O.); mfukuda@yamaguchi-u.ac.jp (M.F.); avivartiya@gmail.com (T.K.); shohei_fujii119459@yahoo.co.jp (S.F.); r.trout.hisa0517@gmail.com (M.H.); uchitomo0513@yahoo.co.jp (T.U.); t-okamu@yamaguchi-u.ac.jp (T.O.); skoba@yamaguchi-u.ac.jp (S.K.); yanoma@yamaguchi-u.ac.jp (M.Y.); 2Department of Cardiology, Ube-Kohsan Central Hospital, Ube 755-0151, Japan; ashimizu@yamaguchi-u.ac.jp; 3Department of Medicine and Clinical Science, Faculty of Health Sciences, Yamaguchi University Graduate School of Medicine, Ube 755-8505, Japan; ta.omuro@gmail.com

**Keywords:** atrial high-rate episode, major adverse cardiovascular event, cardiac implantable electronic device, prognostic value, long-duration AHRE

## Abstract

Long-duration atrial high-rate episodes (AHREs) monitored using cardiac implantable electronic devices (CIEDs) can predict long-term major adverse cardiovascular events (MACEs). This study aimed to compare the impact of long-duration AHRE on MACE development between patients with and without a history of atrial fibrillation (AF). This single-center observational study included 132 CIED-implanted patients with AHREs detected via remote monitoring. The population was dichotomized into groups: with (*n* = 69) and without (*n* = 63) AF. In each group, cumulative incidences of MACEs comprising all-cause deaths, heart failure hospitalizations, strokes, and acute coronary syndromes were compared between patients with AHRE durations of ≥24 h and <24 h. Multivariate analysis was performed to identify predictors of MACEs among patients without AF. MACE incidence was significantly higher in patients with AHRE ≥24 h than in those with <24 h in the group without AF (92% vs. 30%, *p* = 0.005). MACE incidence did not significantly differ between AHRE ≥24 h and <24 h in the group with AF (54% vs. 26%, *p* = 0.44). After a multivariate adjustment, AHRE duration of ≥24 h emerged as the only independent predictor of MACEs among patients without AF (*p* = 0.03). In conclusion, a long-duration AHRE was prognostic in patients without a history of AF but not in patients with a history of AHREs.

## 1. Introduction

The prevalence of atrial fibrillation (AF) is rapidly increasing in developed countries [1]. AF causes stroke and various events, such as heart failure, cardiovascular events, and sudden cardiac death. Hence, AF development can have a negative prognostic impact on mortality [2,3,4]. Previous studies revealed that a high burden of atrial high-rate episodes (AHREs) that were monitored using cardiac implantable electronic devices (CIEDs) could predict future stroke or thromboembolic events, irrespective of the history of AF [5,6,7]. Moreover, recent studies reported that an AHRE is also associated with adverse cardiovascular events, such as heart failure hospitalization (HFH) and cardiovascular events [8,9,10]. Although long-duration AHREs could have a strong impact on these events [9], it remains unclear whether the impact of the long-duration AHREs for such events is equivalent for patients with a history of AF and those without. One might infer that long-duration AHREs for patients without a history of AF could have a stronger prognostic impact than those for patients with a history of AF because appropriate care for an AHRE (e.g., anticoagulation therapy and catheter ablation) might be delayed for the former. To clarify this question, we aimed to compare the impact of long-duration AHREs, which were precisely tracked by remote monitoring, in the development of major adverse cardiovascular events (MACEs) between patients with/without a history of AF. In addition, we aimed to identify predictors of MACE development in patients without a history of AF.

## 2. Materials and Methods

### 2.1. Study Design

This single-center retrospective observational study was conducted at the Yamaguchi University Hospital. The institutional review board approved this study. Informed consent was waived because of the opt-out system. The tenets of the Declaration of Helsinki and the ethical standards of the responsible committee on human experimentation were followed.

Among the consecutive patients who underwent remote monitoring between January 2010 and December 2020 with CIEDs manufactured by four vendors (Medtronic, Inc. (Minneapolis, MN, USA), Boston Scientific (Saint Paul, MN, USA), Biotronik (Berlin, Germany), and Abbott, (Sylmar, CA, USA)), patients in whom AHREs were detected were enrolled. Patients with subcutaneous cardioverter defibrillators and implantable loop recorders were excluded because the algorithm of AHRE detection was not based on an intracardiac electrogram. Patients for which information of AHREs was unavailable and detected AHREs was noise or artifacts were excluded. To evaluate the long-term prognosis, patients in whom the duration of remote monitoring was less than 6 months were excluded. Furthermore, patients for which the AHRE burden had already achieved 100% (persistent AF) in the initiation of remote monitoring were also excluded because the present study aimed to evaluate the development of the AHRE burden. The enrolled patients were allocated into two groups: (I) patients for which AF or atrial flutter had never been documented before the initiation of remote monitoring and (II) patients with a documented history of AF or atrial flutter. In each group, we subcategorized patients based on whether the maximum AHRE duration was ≥24 h or <24 h during the follow-up period. Hence, our previous study showed that the incidence of both ischemic and bleeding events was higher for an AHRE duration of ≥24 h compared to <24 h in patients with CIEDs [7]. We set the cut-off of AHREs as ≥24 h because we hypothesized that prognostic events would also be higher in such patients. We compared the cumulative incidence of MACEs in each group. In the group without a history of AF, we performed univariate and multivariate analyses to identify predictors of MACE development.

### 2.2. Study Endpoints

The primary endpoint of this study was a comparison of the incidence of MACE between patients with AHRE durations of ≥24 h and <24 h in each group (patients with or without a history of AF). In addition, the predictors of MACE development were evaluated by conducting univariate and multivariate analyses in the group without a history of AF.

### 2.3. Definition of Clinical Events

A MACE was defined as a composite of all-cause death, HFH, stroke, and acute coronary syndrome (ACS). HFH was defined as hospitalization that required unplanned medical treatments, such as intravenous administration of diuretics and renal replacement therapy for decompensated heart failure. Stroke was defined as symptomatic ischemic stroke identified by imaging modalities. ACS was defined by the third universal definition of myocardial infarction [11].

### 2.4. Detection of AHRE

An AHRE was defined as an atrial arrhythmia lasting more than 30 s. The cut-off rate was more than 175 bpm for Medtronic, 200 bpm for Biotronik, 170 bpm for Boston Scientific, and 180 bpm for Abbott. The AHRE information was obtained for every transmission.

### 2.5. Remote Monitoring

Between January 2010 and December 2012, patients with a CIED were recommended to undergo remote monitoring at the discretion of the attending cardiologists. Since January 2013, all patients who underwent CIED-related procedures were routinely recommended for the implementation of remote monitoring. In the remote monitoring systems provided by three venders (Home Monitoring™, Biotronik; Latitude Patient Management system™, Boston Scientific, and Merlin.net™, Abbott) the records were automatically transmitted to our institution once a week via home transmitters. In the system from the fourth vendor (CareLink Network™, Medtronic, Inc.), data from the patients with a pacemaker were transmitted once a month, whereas data from patients with an implantable cardioverter defibrillator and cardiac resynchronization therapy were transmitted once a week. The data were reviewed by experienced clinical engineers and electrophysiologists once a week. When unscheduled transmissions were obtained, the data were quickly reviewed. Inquiry for patients was performed by physicians when the scheduled transmission was missed for a week.

### 2.6. Data Collection and Patient Follow-Up

Information on drug and clinical data, including CHA_2_DS_2_-VASc and HAS-BLED scores [1], was collected at the time of initiation of remote monitoring. In the outpatient setting, all patients were followed up once every 6 months and interrogated for data related to the CIEDs. When an AHRE was detected in patients without a history of AF, we recommended that the patients visit primary care physicians and undergo electrocardiographic evaluation. The initiation of anticoagulation therapy was recommended if AF was confirmed in patients with a high thromboembolic risk. Data regarding clinical events were collected by contacting the primary care physician of each patient in July 2021.

### 2.7. Statistical Analysis

Normally distributed variables are expressed as mean ± standard deviation, whereas non-normally distributed variables are expressed as medians and interquartile ranges (first and third quartiles). Differences in continuous variables between patients with AHRE durations of ≥24 h and <24 h were evaluated using the Mann–Whitney U test. Categorical variables are presented as frequencies and proportions (%) and were compared using the chi-square (χ^2^) test. The differences in the cumulative incidence of the first MACE and each clinical event were compared using the log-rank test. Cox proportional hazards regression analysis was performed to identify the predictors of MACEs in patients without a history of AF. Variables with *p*-values ≤ 0.1 in the univariate analysis were selected as potential predictors. To precisely evaluate the association between long-duration AHREs and MACEs, a sensitivity analysis of the log-rank test was performed for the population by excluding patients who developed a MACE before the AHRE duration exceeded the 24 h threshold. In addition, to evaluate the association between anticoagulation therapy and MACE incidence in the group without a history of AF, we performed a sensitivity analysis of the log-rank test performed for the population by excluding patients who were prescribed anticoagulants for indications other than AF. The results are expressed as hazard ratios and 95% confidence intervals (CIs). All analyses were performed using SPSS version 19 (IBM Corp., Armonk, NY, USA), and results with a *p*-value < 0.05 were considered statistically significant.

## 3. Results

### 3.1. Study Population

A flow diagram of the present study is shown in Figure 1. During the entire follow-up period, 960 patients underwent CIED monitoring. Of these, 550 patients were monitored with CIEDs manufactured by the four vendors who agreed to implement remote monitoring. In the present study, 132 patients were included. The median follow-up period of patients in whom AHREs were detected was 4.6 (2.2, 7.3) years (634 person-years). The data were transmitted 133 (63, 293) times during 4.4 (2.2, 7.3) years of remote monitoring. The proportion of each vendor was 39% for Medtronic, 27% for Boston Scientific, 26% for Biotronik, and 8% for Abbott. Sixty-three patients without a history of AF developed AHREs at 1.2 (0.2, 2.7) years since the initiation of remote monitoring. Twenty-two patients (35%) had a maximum AHRE duration of ≥24 h. Of those, 54% (12/22 patients) achieved ≥24 h duration in the first AHRE event. The demographic characteristics of patients with an AHRE duration of ≥24 h or <24 h are compared in Table 1. Overall, 22% of patients were prescribed anticoagulants but not diagnosed with AF. The indications were the prevention of intramural thrombosis (64%), followed by deep vein thrombosis (29%) and venous graft thrombosis (7%). In the group with a history of AF, 37 patients (54%) had a maximum AHRE duration of ≥24 h at 0.7 (0.1, 2.1) years since the initiation of remote monitoring. The demographics of patients in this group with AHRE durations of ≥24 h or <24 h are compared in Table 2.

### 3.2. Cumulative Incidence of MACEs

Figure 2A and Table 3a show a comparison of the cumulative incidence of MACEs between patients with AHRE ≥24 h and <24 h in the group without a history of AF. In total, 24 patients (8/100 person-years) developed at least one clinical event (Appendix A). The cumulative incidence of MACEs was significantly higher in patients with AHRE ≥24 h than in those with AHRE <24 h (92% (95% CI: 78–100%), 14/100 person-years vs. 30% (95% CI: 11–49%), 4/100 person-years, *p* = 0.005). The cumulative incidence of each clinical event is shown in Figure 3. The cumulative incidence of HFHs was significantly higher in patients with AHRE ≥24 h than in those with AHRE <24 h (72% (95% CI: 48–97%), 10/100 person-years vs. 30% (95% CI: 0–70%), 1/100 person-years, *p* = 0.0003). This trend persisted in the sensitivity analysis for MACE (89% (95% CI: 70–100%) vs. 30% (95% CI: 11–49%), *p* = 0.08). The sensitivity analysis for HFHs showed a statistically significant difference persisted (62% (95% CI: 36–88%) vs. 30% (95% CI: 0–70%), *p* = 0.006). Regarding another sensitivity analysis, the cumulative incidence of MACEs was significantly higher in patients with AHRE ≥24 h than in those with AHRE <24 h, even in the population which excluded the patients who were prescribed anticoagulants for indications other than AF (69% (52–93%) vs. 12% (2–66%), *p* = 0.02). In the group with a history of AF, 21 patients (6/100 person-years) developed at least one clinical event. A comparison of the cumulative incidence of MACEs and each clinical event in the group with a history of AF is shown in Figure 2B, Figure 4 and Table 3b. Interestingly, there were no differences in MACEs and HFHs between patients with AHRE ≥24 h and those with AHRE <24 h (MACE: 54% (95% CI: 29–79%), 9/100 person-years vs. 26% (95% CI: 9–43%), 4/100 person-years, *p* = 0.44; HFH: 46% (95% CI: 22–71%), 7/100 person-years vs. 14% (95% CI: 1–27%), 1/100 person-years, *p* = 0.12).

### 3.3. Catheter Ablation during the Follow-Up Period

During the follow-up period, 5% (3/63) of patients without a history of AF and 27% (19/69) of patients with a history of AF underwent catheter ablation for atrial tachyarrhythmia, indicating that the proportion of patients who underwent catheter ablation was significantly smaller in the group without a history of AF than in the group with a history of AF (*p* = 0.0005). Among patients who developed MACEs, 20% of patients (9/45 patients) underwent catheter ablation. In the group without a history of AF, all patients underwent catheter ablation following the development of the first MACE (2/2 patients, 100%). Even in the group with a history of AF, more than half of the patients underwent catheter ablation following the development of the first MACE (4/7 patients, 57%).

### 3.4. Anticoagulation Therapy for Patients Who Developed MACEs

In the entire population, the proportion of patients who were prescribed anticoagulants during the entire follow-up period was also significantly smaller in the group without a history of AF than in those with a history of AF (29/63 patients, 46% vs. 69/69 patients, 100%, *p* < 0.0001). In patients who developed MACEs, most patients (40/45 patients, 89%) received anticoagulation therapy during the entire follow-up period. However, the prevalence of patients who were prescribed anticoagulants prior to the first MACE was significantly lower in the group without a history of AF than in the group with a history of AF (14/24 patients, 58% vs. 20/21 patients, 95%; *p* = 0.004).

### 3.5. Predictors of MACE Development

Results of the univariate and multivariate analyses performed to evaluate the predictors of MACE development in the group without a history of AF are summarized in Table 4. Age >75 years, AHRE duration ≥24 h, left ventricular ejection fraction (LVEF) <40%, and brain natriuretic peptide level >200 pg/mL emerged as significant factors in the univariate analysis. Among these variables, AHRE duration ≥24 h was the only independent predictor after a multivariate adjustment.

## 4. Discussion

### 4.1. Main Findings

The important findings of this study were as follows: First, in the group without a history of AF, the cumulative incidence of MACEs was significantly higher in patients with AHRE ≥24 h than in those with <24 h. The gap was mainly derived from the difference in HFH. However, there was no difference in MACE and HFH incidences between patients with AHRE ≥24 h and those with <24 h in the group with a history of AF (Graphical Abstract). Second, AHRE duration ≥24 h was the only independent predictor of MACEs among patients without a history of AF after a multivariate adjustment.

### 4.2. The Impact of Long Duration of AHRE for MACE

Several studies showed the association between AHREs and MACEs in patients with CIEDs [8,9,10,12,13,14]. In general, previous studies commonly reported that patients with AHREs were more susceptible to developing MACEs, such as HFH and cardiovascular death, than those without AHRE [12,13,14]. However, studies focusing on the association between the duration of AHRE burden (i.e., short vs. long duration) and the risk of a MACE are limited [9,10]. Liu et al. reported that patients with an AHRE duration of ≥6 min had a higher risk of a MACE (mainly composed of ACS) than those with a shorter (<6 min) AHRE, particularly when the patients had a history of AF or myocardial infarction [9]. However, the study could not show a linear relationship between a longer duration of an AHRE and a higher risk of a MACE. In contrast, Pastori et al. showed that patients who developed a long-duration (≥24 h) AHRE had nearly a two-fold higher risk of a MACE than those who developed a short-duration (≥5 min) AHRE [10]. In agreement with the findings of these studies, our results also indicated a higher incidence of MACEs, along with a longer duration of AHRE in patients without a history of AF. Even in a study investigating the incidence of HFHs in patients with CIEDs, Nishinarita et al. revealed that the risk was higher in the group with a high AHRE burden than in the groups with low or no AHREs [8]. Hence, given that the most common event in our population was an HFH, which is similar to that in the population of the studies [8,10], we assumed that there would be a linear relationship between AHRE duration and HFH incidence. Our data added evidence showing that the risk of HFH was higher in a long-duration AHRE than in a short-duration AHRE.

### 4.3. Difference in the Impact of AHRE between Patients with and without a History of AF

To the best of our knowledge, this is the first study to demonstrate the difference in the prognostic impact of long-duration AHRE in patients with and without a history of AF. Rapid AF induced loss of cardiac output owing to several mechanisms, such as a loss of atrial kick, irregularity of ventricular systole, and incomplete left ventricular relaxation owing to tachycardia [15,16]. Notably, it would be challenging to maintain hemodynamic compensation for new-onset AF. Previous studies showed that new-onset AF has a higher prognostic impact than prevalent AF in patients with chronic heart failure and acute myocardial infarction [17,18]. This phenomenon could also reflect the fact that the therapeutic management of new-onset AF tended to be incomplete and delayed. Hence, we postulated that our results would also follow this reasoning. To support this assumption, our population showed that the number of patients who underwent catheter ablation in the group without a history of AF was significantly fewer than in the group with AF. Even with anticoagulation therapy, significantly fewer patients without AF than those with AF who developed MACEs were prescribed anticoagulants prior to the first MACE. Hence, our observation also suggested that therapeutic management of AHREs, as well as early detection, might be mandatory to prevent MACEs. Since our population was composed of patients who underwent remote monitoring, we could obtain information regarding AHRE earlier than that in other studies. Although remote monitoring allowed for AHRE detection several months earlier than an in-person follow-up [19], our results indicated that early detection itself might be insufficient to reduce MACEs. Currently, there are several therapeutic options for AHRE. The latest guideline states that patients who were detected with long-duration AHRE (>24 h) without the diagnosis of AF should be considered for anticoagulation therapy if they have a high thromboembolic risk [1]. Atrial tachycardia pacing could help prevent prolongation of AHRE duration [20]. Catheter ablation for AF could improve the prognostic outcome in patients with systolic impairment and clinically confirmed AF [21]. Early implementation of such therapeutic options for AHRE might be helpful in improving the clinical outcomes in patients without a history of AF.

### 4.4. Clinical Implications

Our study revealed that patients without a history of AF had poorer prognostic outcomes when they were detected with long-duration AHRE than with short-duration AHRE. Thus, our results provide information regarding patients at high risk for MACEs (i.e., those with AHRE ≥24 h and without a history of AF). In addition, therapeutic management for AHRE could be delayed in patients without a history of AF, unlike in patients previously diagnosed with AF. Hence, our results might encourage clinicians to initiate early therapies in patients without a history of AF who experience long-duration AHRE. Regarding newly diagnosed AF, a recent trial showed that an early rhythm control strategy could improve the prognostic outcome compared to usual care [22]. Early therapeutic strategies for AHREs may also be helpful in improving MACEs.

### 4.5. Limitations

The present study had several limitations. First, the study protocol was designed as a single-center, retrospective observational study. Thus, we could have missed unmeasured variables associated with MACEs. In addition, the sample size was relatively small. Hence, it remains unclear whether our results can be extrapolated to other populations; further studies in other institutions are needed to validate our findings. Second, the present data were composed of a selected population in which patients who were recommended for implementation of remote monitoring were enrolled. Thus, our data included patients with high severity. It remains uncertain whether our findings could be applicable to the general population that comprises patients with usual follow-up regimens. Third, a certain proportion of patients in our population were prescribed anticoagulants, although they were not diagnosed with AF. This might reflect that the incidence of stroke was lower than other events, such as all-cause death and HFH, in our population. Hence, the incidence of stroke varied in response to the proportion of patients prescribed with anticoagulants, and it remains unclear whether our results could be extrapolated to a population wherein only a few patients were prescribed anticoagulants. Fourth, our study design could not differentiate whether an AHRE was the causal factor for developing a MACE or the result of the progression of other comorbidities. Hence, it remained unclear whether aggressive therapeutic management for AHREs might contribute to the reduction of MACEs. Further studies comparing the temporal relationship between the duration of AHREs and other factors, such as echocardiographic parameters and biomarkers, in individual case levels could clarify the issue. However, we believe it is important for clinicians to be cautious about the duration of AHRE because it can be a useful marker to identify patients with high risk for developing MACE.

## 5. Conclusions

Our data demonstrated that a long-duration AHRE in patients without a history of AF could be highly associated with prognostic events than in patients with a history of AF. Our results suggested that the risk of a MACE, especially HFH, could be quite high when patients without a history of AF experienced a long-duration AHRE.

## Figures and Tables

**Figure 1 jcm-11-01732-f001:**
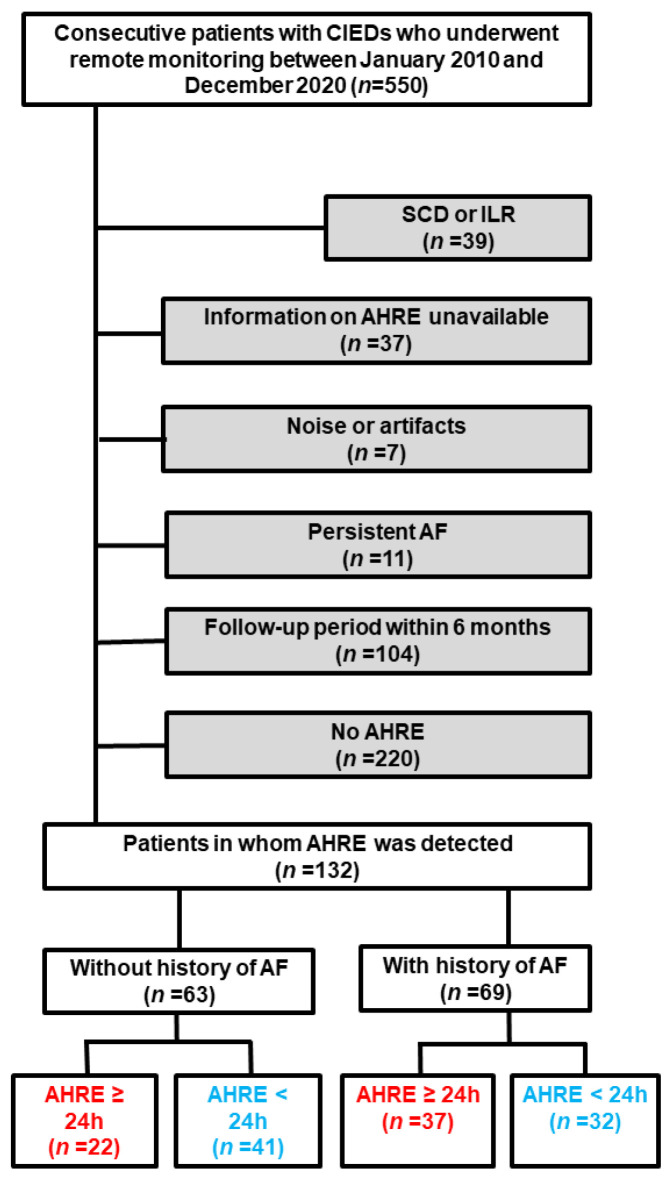
Flow diagram of the study. Red and blue texts indicated AHRE of ≥24 h and <24 h, respectively. Grey background indicated excluded patients. Blue texts indi AF, atrial fibrillation; AHRE, atrial high-rate episodes; CIED, cardiac implantable electronic device; ILR, implantable loop recorder; SCD, subcutaneous cardioverter-defibrillator.

**Figure 2 jcm-11-01732-f002:**
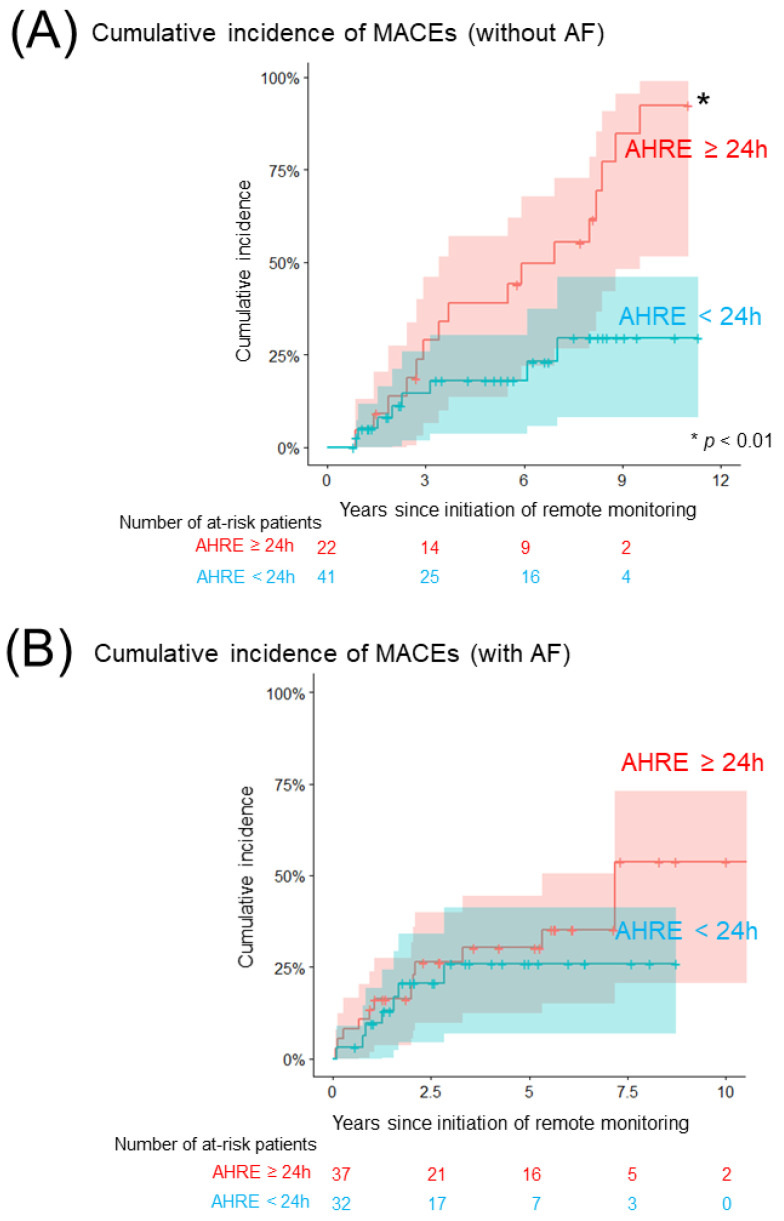
Comparison of the cumulative incidence of MACEs. (**A**) The Kaplan–Meier curve shows the cumulative incidence and 95% CI of MACEs following the initiation of remote monitoring among patients without a history of AF (red: patients with AHRE ≥24 h, blue: <24 h). (**B**) The Kaplan–Meier curve shows the cumulative incidence and 95% CI of MACEs following the initiation of remote monitoring among patients with a history of AF (red: patients with AHRE ≥24 h, blue: <24 h). The asterisk indicates statistical significance (* *p* < 0.01). AF, atrial fibrillation; AHRE, atrial high-rate episode; CI, confidence interval; MACE, major adverse cardiovascular event.

**Figure 3 jcm-11-01732-f003:**
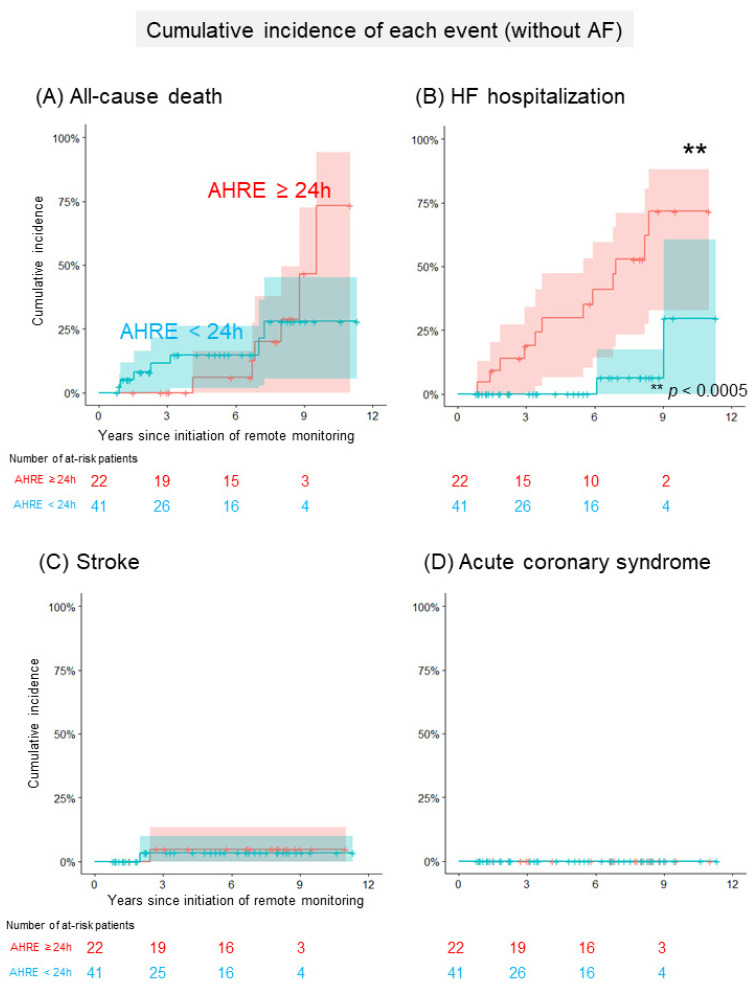
Comparison of the cumulative incidence of each event in patients without a history of AF. (**A**) The Kaplan–Meier curve shows the cumulative incidence and 95% CI of all-cause deaths following the initiation of remote monitoring in patients without a history of AF (red: patients with AHRE ≥24 h, blue: <24 h). (**B**) The Kaplan–Meier curve shows the cumulative incidence and 95% CI of heart failure hospitalizations following the initiation of remote monitoring in patients without a history of AF (red: patients with AHRE ≥24 h, blue: <24 h). (**C**) The Kaplan–Meier curve shows the cumulative incidence and 95% CI of strokes following the initiation of remote monitoring in patients without a history of AF (red: patients with AHRE ≥24 h, blue: <24 h). (**D**) The Kaplan–Meier curve shows the cumulative incidence and 95% CI of acute coronary syndromes following the initiation of remote monitoring in patients without AF (red: patients with AHRE ≥24 h, blue: <24 h). The asterisks indicate statistical significance (** *p* < 0.0005). AF, atrial fibrillation; AHRE, atrial high-rate episode; CI, confidence interval.

**Figure 4 jcm-11-01732-f004:**
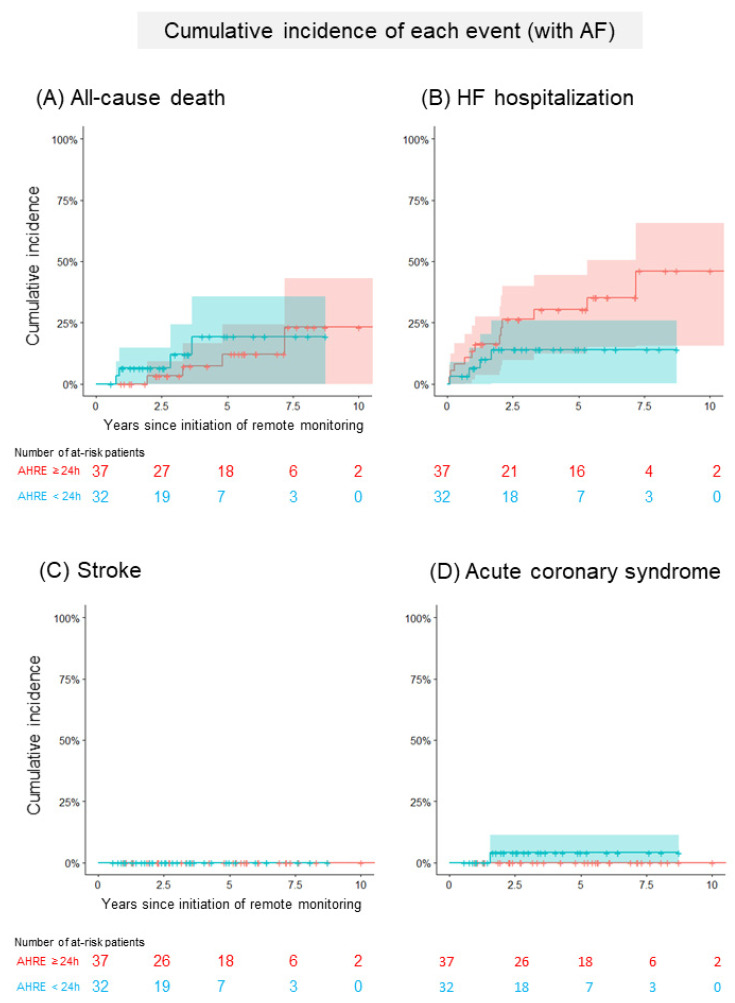
Comparison of the cumulative incidence of each event in patients with a history of AF. (**A**) The Kaplan–Meier curve shows the cumulative incidence and 95% CI of all-cause deaths following the initiation of remote monitoring in patients with a history of AF (red: patients with AHRE ≥24 h, blue: <24 h). (**B**) The Kaplan–Meier curve shows the cumulative incidence and 95% CI of heart failure hospitalizations following the initiation of remote monitoring in patients with a history of AF (red: patients with AHRE ≥24 h, blue: <24 h). (**C**) The Kaplan–Meier curve shows the cumulative incidence and 95% CI of strokes following the initiation of remote monitoring in patients with a history of AF (red: patients with AHRE ≥24 h, blue: <24 h). (**D**) The Kaplan–Meier curve shows the cumulative incidence and 95% CI of acute coronary syndromes following the initiation of remote monitoring in patients with a history of AF (red: patients with AHRE ≥24 h, blue: <24 h). AF, atrial fibrillation; AHRE, atrial high-rate episodes; CI, confidence interval.

**Table 1 jcm-11-01732-t001:** Comparison of demographics between patients with AHRE ≥24 h and <24 h in patients without a history of AF.

	Total (*n* = 63)	AHRE ≥24 h (*n* = 22)	AHRE <24 h (*n* = 41)	*p*-Value
Age (years), mean ± SD	68 ± 16	67 ± 14	69 ± 16	0.51
Male sex, *n* (%)	45 (71)	18 (82)	27 (66)	0.18
BW (kg), mean ± SD	60 ± 10	63 ± 9	59 ± 10	0.14
^†^ ICD/CRT, *n* (%)	35 (56)	16 (73)	19 (46)	0.04
SBP (mmHg), mean ± SD	127 ± 28	118 ± 21	132 ± 30	0.06
HR (min^−1^), mean ± SD	66 ± 12	67 ± 10	65 ± 13	0.51
NYHA class, mean ± SD	1.4 ± 0.6	1.6 ± 0.6	1.4 ± 0.6	0.1
History of HFH, *n* (%)	9 (14)	5 (23)	4 (10)	0.16
CHA_2_DS_2_-VASc, mean ± SD	3 ± 1.8	2.9 ± 1.8	3 ± 1.8	0.78
HAS-BLED, mean ± SD	1.4 ± 1.2	1.4 ± 1	1.5 ± 1.2	0.85
Etiology of SHD				
CAD, *n* (%)	9 (14)	5 (23)	4 (10)	0.16
DCM/DHCM, *n* (%)	8 (13)	2 (9)	6 (15)	0.52
HCM, *n* (%)	12 (19)	6 (27)	6 (15)	0.22
VHD, *n* (%)	3 (5)	1 (5)	2 (5)	0.95
CHD, *n* (%)	4 (6)	3 (14)	1 (2)	0.08
Echocardiographic parameter				
^†^ LVDd (mm), mean ± SD	53 ± 11	56 ± 11	51 ± 11	0.03
LVEF (%), mean ± SD	55 ± 18	51 ± 19	57 ± 18	0.16
LAD (mm), mean ± SD	42 ± 7	43 ± 9	41 ± 6	0.14
Therapeutic agent				
ACEI/ARB, *n* (%)	37 (59)	15 (68)	22 (54)	0.26
Beta blocker, *n* (%)	36 (57)	15 (68)	21 (51)	0.19
MRA, *n* (%)	19 (30)	6 (27)	13 (31)	0.71
Diuretics, *n* (%)	24 (38)	10 (45)	14 (34)	0.38
Amiodarone, *n* (%)	15 (24)	8 (36)	7 (17)	0.09
VKA/DOAC, *n* (%)	14 (22)	7 (32)	7 (17)	0.18
Laboratory data				
eGFR (mL/min/1.73 m^2^), mean ± SD	59 ± 23	54 ± 20	62 ± 24	0.17
BNP level (pg/mL), median (IQR)	183 (72, 379)	258 (72, 417)	131 (64, 372)	0.29

Numerical data are expressed as mean ± SD or median (interquartile range (IQR); first quartile, third quartile). Categorical data are expressed as percentages and numbers. ^†^ indicates statistical significance (*p* < 0.05). ACEI, angiotensin-converting enzyme inhibitor; AF, atrial fibrillation; AHRE, atrial high-rate episode; ARB, angiotensin II receptor blocker; BNP, brain natriuretic peptide; BW, body weight; CAD, coronary artery disease; CHD, congenital heart disease; CRT, cardiac resynchronization therapy; DCM, dilated cardiomyopathy; DHCM, dilated phase of hypertrophic cardiomyopathy; DOAC, direct oral anticoagulant; eGFR, estimated glomerular filtration rate; HCM, hypertrophic cardiomyopathy; HFH, heart failure hospitalization; HR, heart rate; ICD, implantable cardioverter-defibrillator; IQR, interquartile range; LAD, left atrial diameter; LVDd, left ventricular end-diastolic diameter; LVEF, left ventricular ejection fraction; MRA, mineralocorticoid receptor antagonist; NYHA, New York Heart Association; SBP, systolic blood pressure; SD, standard deviation; SHD, structural heart disease; VHD, valvular heart disease; VKA, vitamin K antagonist.

**Table 2 jcm-11-01732-t002:** Comparison of demographics between patients with AHRE ≥24 h and <24 h in patients with a history of AF.

	Total (*n* = 69)	AHRE ≥24 h (*n* = 37)	AHRE <24 h (*n* = 32)	*p*-Value
Age (years), mean ± SD	71 ± 11	71 ± 11	70 ± 12	0.57
Male sex, *n* (%)	43 (62)	22 (59)	21 (66)	0.6
BW (kg), mean ± SD	58 ± 12	56 ± 11	59 ± 13	0.38
ICD/CRT, *n* (%)	31 (45)	19 (51)	12 (38)	0.25
SBP (mmHg), mean ± SD	124 ± 17	123 ± 17	125 ± 17	0.38
HR (min^−1^), mean ± SD	67 ± 12	69 ± 11	65 ± 14	0.1
NYHA class, mean ± SD	1.5 ± 0.7	1.6 ± 0.7	1.4 ± 0.7	0.24
History of HFH, *n* (%)	14 (20)	8 (22)	6 (19)	0.76
CHA_2_DS_2_-VASc, mean ± SD	3.5 ± 1.7	3.8 ± 1.8	3.1 ± 1.6	0.14
HAS-BLED, mean ± SD	1.9 ± 1.4	2.3 ± 1.6	1.5 ± 1	0.06
Etiology of SHD				
CAD, *n* (%)	11 (16)	8 (22)	3 (9)	0.16
DCM/DHCM, *n* (%)	2 (3)	2 (5)	0	0.18
HCM, *n* (%)	10 (14)	6 (16)	4 (12)	0.66
VHD, *n* (%)	7 (10)	6 (16)	1 (3)	0.07
CHD, *n* (%)	2 (3)	1 (3)	1 (3)	0.91
Echocardiographic parameter				
LVDd (mm), mean ± SD	50 ± 7	50 ± 8	49 ± 6	0.79
LVEF (%), mean ± SD	56 ± 14	54 ± 14	58 ± 13	0.14
LAD (mm), mean ± SD	43 ± 7	44 ± 7	43 ± 7	0.55
Therapeutic agent				
ACEI/ARB, *n* (%)	46 (67)	24 (65)	22 (69)	0.59
Beta blocker, *n* (%)	43 (62)	21 (57)	22 (69)	0.22
MRA, *n* (%)	15 (22)	9 (24)	6 (19)	0.62
Diuretics, *n* (%)	30 (43)	17 (46)	13 (41)	0.74
Amiodarone, *n* (%)	15 (22)	11 (30)	4 (13)	0.09
VKA/DOAC, *n* (%)	66 (96)	34 (92)	32 (100)	0.1
Laboratory data				
eGFR (mL/min/1.73 m^2^), mean ± SD	52 ± 20	49 ± 21	55 ± 19	0.3
^†^ BNP level (pg/mL), median (IQR)	138 (57, 315)	172 (102, 382)	90 (40, 216)	0.04

Numerical data are expressed as mean ± SD or median (interquartile range (IQR); first quartile, third quartile). Categorical data are expressed as percentages and numbers. ^†^ indicates statistical significance (*p* < 0.05). ACEI, angiotensin-converting enzyme inhibitor; AF, atrial fibrillation; AHRE, atrial high-rate episode; ARB, angiotensin II receptor blocker; BNP, brain natriuretic peptide; BW, body weight; CAD, coronary artery disease; CHD, congenital heart disease; CRT, cardiac resynchronization therapy; DCM, dilated cardiomyopathy; DHCM, dilated phase of hypertrophic cardiomyopathy; DOAC, direct oral anticoagulant; eGFR, estimated glomerular filtration rate; HCM, hypertrophic cardiomyopathy; HFH, heart failure hospitalization; HR, heart rate; ICD, implantable cardioverter-defibrillator; IQR, interquartile range; LAD, left atrial diameter; LVDd, left ventricular end-diastolic diameter; LVEF, left ventricular ejection fraction; MRA, mineralocorticoid receptor antagonist; NYHA, New York Heart Association; SBP, systolic blood pressure; SD, standard deviation; SHD, structural heart disease; VHD, valvular heart disease; VKA, vitamin K antagonist.

**Table 3 jcm-11-01732-t003:** The cumulative incidence of each clinical event in patients without (a)/with (b) a history of AF.

**(a) Without a History of AF**			
	**AHRE ≥24 h**	**AHRE <24 h**	***p*-Value**
^†^ MACE, % (95% CI)	92 (78, 100)	30 (11, 49)	0.005
All-cause death, % (95% CI)	73 (33, 100)	28 (8, 48)	0.88
^†^ HFH, % (95% CI)	72 (48, 97)	30 (0, 70)	0.0003
Stroke, % (95% CI)	5 (0, 14)	4 (0, 10)	0.86
ACS, % (95% CI)	0	0	N/A
**(b) With a History of AF**			
	**AHRE ≥24 h**	**AHRE <24 h**	***p*-Value**
MACE, % (95% CI)	54 (29, 79)	26 (9, 43)	0.44
All-cause death, % (95% CI)	24 (1, 47)	19 (1, 38)	0.51
HFH, % (95% CI)	46 (22, 71)	14 (1, 27)	0.12
Stroke, % (95% CI)	0	0	N/A
ACS, % (95% CI)	0	4 (0, 12)	0.25

ACS, acute coronary syndrome; AF, atrial fibrillation; CI, confidence interval; HFH, heart failure hospitalization; MACE, major adverse cardiovascular event; N/A, not available. ^†^ denotes a statistically significant finding.

**Table 4 jcm-11-01732-t004:** Identification of predictive factors for MACEs in patients without a history of AF.

	Univariate Analysis			Multivariate Analysis		
	HR	95% CI	*p*-Value	HR	95% CI	*p*-Value
Age >75 years	2.1	0.92–4.8	0.08	1.5	0.61–3.4	0.39
Male sex	2.2	0.77–6.5	0.14			
^†^ AHRE ≥24 h	3.2	1.3–7.4	0.007	3.0	1.1–8.1	0.03
NYHA >2	3.8	0.48–31	0.2			
CHA_2_DS_2_-VASc ≥3	1.3	0.59–3	0.48			
HAS-BLED ≥3	1.6	0.5–4.8	0.41			
LVDd >55 mm	1.5	0.63–3.7	0.35			
VKA/DOAC	1.5	0.6-3.8	0.37			
LVEF <40%	2.3	0.9–5.9	0.08	1.6	0.52–4.7	0.42
LAD >45 mm	1.7	0.77–3.8	0.18			
eGFR <30 mL/min/1.73 m^2^	1.9	0.84–4.6	0.11			
BNP level >200 pg/mL	2.7	1.1–6.3	0.02	2.1	0.74–6.1	0.16

^†^ indicates statistical significance after adjustment in the multivariate analysis (*p* < 0.05). HR, hazard ratio; AHRE, atrial high-rate episode; BNP, brain natriuretic peptide; DOAC, direct oral anticoagulants; eGFR, estimated glomerular filtration rate; LAD, left atrial diameter; LVDd, left ventricular end-diastolic diameter; LVEF, left ventricular ejection fraction; NYHA, New York Heart Association; VKA, vitamin K antagonist.

## Data Availability

The data sets analyzed in this study are available from the corresponding author upon reasonable request.

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
