# Peer review of "The Differential Prognostic Impact of Long-Duration Atrial High-Rate Episodes Detected by Cardiac Implantable Electronic Devices between Patients with and without a History of Atrial Fibrillation"

_jcm, 2022, doi:10.3390/jcm11061732_

Round 1
Reviewer 1 Report
Congratulations on this well written piece and thank you for letting me share my thoughts. Please find below my comments:
- Reading your manuscript, the use of anticoagulants seems to be of great importance regarding potential development of MACE. You primarily compare patients with history of AF, to patients without history of AF. a) Why not compare patients with concurrent anticoagulant use, to patients without? b) Is there any literature available on the use of anticoagulant therapy and the development of MACE? if so, this might be interesting to include in the paper and I'd encourage you to do so.
- Why did you choose for a cut-off of 24h, when it comes to defining long or short AHRE duration? Might add a reference to explain this.
- The median follow-up period of patients in whom AHRE was detected was 4.6 years (634 person-years). Is this long enough for the development of MACE? Could we have missed any MACE cases due to this?
- Figure 1 requires formatting, the flowchart fills up the entire page, might be better to reduce its size.
- Figure 2A shows the Kaplan-Meier of the cumulative incidence of MACE in patients without history of AF. There are patients in this group that are using anticoagulants for any other medical reason. Would it be interesting to leave these patients out and see what the data and the Kaplan-Meier look like then? So solely focusing on patients not on anticoagulant therapy: are these the patients that develop MACE?
- Your findings suggest a higher incidence of MACE in patients with a longer duration of AHRE (in patients without AF history): could you try to elaborate on the potential (pathofysiologic) mechanism behind this? Why is this the case?
- In the conclusions section, you state "Our data demonstrated .... etc." I think this statement is too bold and needs some more nuance, regarding your retrospective study design with limited sample size. Please rephrase accordingly.
Thanks and again my compliments with your work.
Reviewer 2 Report
The authors evaluated the impact of long-duration AHRE on MACE development between patients with and without a history of atrial fibrillation (AF). They concluded that AHRE ≥24 h was prognostic in patients without a history of AF but not in patients with AF.
I have the following concerns:
- Single-center retrospective observational study with 132 patients with confirmed or not confirmed AHRE limits the study.
- The results with values should be included in the Abstract.
- What are the practical implications of the study?
Round 2
Reviewer 2 Report
Thank you. All my concerns have been adequately addressed. I have no further comments.